# *Wnt5a–Vangl1/2* signaling regulates the position and direction of lung branching through the cytoskeleton and focal adhesions

**Kuan Zhang, Erica Yao, Ethan Chuang, Biao Chen, Evelyn Y. Chuang, Regan F. Volk, Katherine L. Hofmann, Balyn Zaro, Pao-Tien Chuang***

Cardiovascular Research Institute, University of California, San Francisco, California, United States of America

\* pao-tien.chuang@ucsf.edu

**Data Availability Statement:** Gene Expression Omnibus database accessible through GEO Series accession number (GSE188993). UCSD Mass

## Abstract

Lung branching morphogenesis requires reciprocal interactions between the epithelium and mesenchyme. How the lung branches are generated at a defined location and projected toward a specific direction remains a major unresolved issue. In this study, we investigated the function of Wnt signaling in lung branching in mice. We discovered that *Wnt5a* in both the epithelium and the mesenchyme plays an essential role in controlling the position and direction of lung branching. The *Wnt5a* signal is mediated by *Vangl1/2* to trigger a cascade of noncanonical or planar cell polarity (PCP) signaling. In response to noncanonical *Wnt* signaling, lung cells undergo cytoskeletal reorganization and change focal adhesions. Perturbed focal adhesions in lung explants are associated with defective branching. Moreover, we observed changes in the shape and orientation of the epithelial sheet and the underlying mesenchymal layer in regions of defective branching in the mutant lungs. Thus, PCP signaling helps define the position and orientation of the lung branches. We propose that mechanical force induced by noncanonical *Wnt* signaling mediates a coordinated alteration in the shape and orientation of a group of epithelial and mesenchymal cells. These results provide a new framework for understanding the molecular mechanisms by which a stereotypic branching pattern is generated.

## Introduction

Branching morphogenesis is a fundamental mechanism for pattern formation [1,2]. It is utilized by many organs and vasculature to generate a defined pattern required for tissue function. The lung, kidney, mammary gland, salivary gland, pancreas, and prostate are among the branching organs that have been extensively studied. For instance, many genes and pathways that control lung branching have been discovered [3–5]. However, we still lack a complete mechanistic understanding of how new lung branches are formed and extended in a spatially and temporally specific manner. In particular, the cellular and molecular basis of how lung epithelial cells undergo morphogenetic changes to produce a new branch remains underexplored [6,7].

Spectrometry Interactive Virtual Environment (MassIVE) number MSV000089851.

**Funding:** PTC was supported by National Institutes of Health R01 HL142876 (National Heart, Lung, and Blood Institute). The funders had no role in study design, data collection and analysis, decision to publish, or preparation of the manuscript.

**Competing interests:** The authors have declared that no competing interests exist.

**Abbreviations:** cRAP, common repository of adventitious proteins; dpc, days post coitus; ECM, extracellular matrix; FAK, focal adhesion kinase; FDR, false discovery rate; FZ, Frizzled; GO, gene ontology; p-Cofilin, phosphorylated Cofilin; PCP, planar cell polarity; PEI, polyethylenimine; PFA, paraformaldehyde; p-FAK, phosphorylated FAK; pMLC, phosphorylated myosin light chain; RAc, right accessory; RCd, right caudal; RCr, right cranial; RMd, right middle.

The planar cell polarity (PCP) pathway is an evolutionarily conserved mechanism for orchestrating cell shape and motility during pattern formation [8–11]. PCP signaling has been broadly investigated. The major components of the PCP pathway are known, and their genetic interactions have been defined. The PCP pathway is the noncanonical branch of the *Wnt* pathway. Similar to the canonical *Wnt* pathway, PCP signaling is triggered by binding of the Wnt ligands to their cell surface receptors that include the Frizzled (FZ) receptors and ROR coreceptors. However, instead of controlling β-catenin levels as seen in the canonical *Wnt* pathway, PCP signaling utilizes multiple transmembrane and cytoplasmic proteins to regulate the actomyosin cytoskeleton. How changes in cellular properties induced by PCP signaling influence branching morphogenesis is a key unresolved question. Insight into this issue will offer a new framework for understanding branching morphogenesis.

Among the several Wnts that are expressed in the lung, *Wnt5a* is a prominent member of the noncanonical Wnt family [12]. *Wnt5a* is also capable of mediating canonical *Wnt* signaling [13,14]. The role of *Wnt5a* in lung branching has not been fully explored. A previous report on *Wnt5a*$^{-/-}$ mouse lungs primarily focused on later stages (e.g., 16.5 to 18.5 days post coitus (dpc)) of lung development and concluded that *Wnt5a* controls distal lung morphogenesis [15]. Whether *Wnt5a* regulates early lung branching is unknown and the sources of *Wnt5a* in this process were not functionally defined.

Similarly, how the downstream effectors of *Wnt5a* control lung branching is unclear. In the literature, a hypomorphic (reduced function) allele of *Vangl2*, *Vangl2*$^{Lp}$ (*loop tail*) [16], has been widely used. *Vangl1* and *Vangl2* encode the mammalian homologs of fly *Van Gogh (Vang)/strabismus* and are absolutely required for PCP signaling [17]. Homozygous *Vangl2*$^{Lp/Lp}$ mice die in utero due to an open neural tube. Analysis of *Vangl2*$^{Lp/Lp}$ lungs revealed defective branching, resulting in fewer branches and narrow lumens [18]. Disrupted cytoskeletal organization was also observed in *Vangl2*$^{Lp/Lp}$ lungs [18]. However, intact *Vangl1* and residual *Vangl2* activity in *Vangl2*$^{Lp/Lp}$ mutants retained PCP signaling and prevented an accurate assessment of how PCP signaling promotes lung branching, especially during the early steps of branch formation. A complete loss of both *Vangl1* and *Vangl2* function is required to uncover the molecular mechanisms by which PCP signaling regulates branching. In addition, loss of PCP signaling in a select compartment (e.g., the lung epithelium or mesenchyme) through conditional gene inactivation is necessary to investigate PCP signaling in different niches.

In this study, we have defined the role of epithelial and mesenchymal *Wnt5a* in controlling the position and direction of lung branching. A complete loss of *Vangl1/2*, the effectors of *Wnt5a*, resulted in similar albeit milder phenotypes than those due to *Wnt5a* removal. We discovered that cytoskeletal reorganization induced by PCP signaling leads to changes in focal adhesions required for branching. This is associated with alterations in the shape and orientation of the epithelial sheet and the underlying mesenchymal layer in regions of defective branching in the mutant lungs. Together, these novel findings reveal a molecular cascade that controls cellular properties required for branching morphogenesis.

## Results

### Global inactivation of *Wnt5a* perturbs the position and direction of early lung branching

To search for signals that trigger PCP signaling and lung branching, we tested the function of *Wnt5a* and examined branching in lungs of *Wnt5a*$^{-/-}$ mice especially at the early stages of lung development. The null allele of *Wnt5a* (*Wnt5a*$^-$) was derived from the floxed allele of *Wnt5a* (*Wnt5a*$^f$) [19] by *Sox2-Cre* [20]. We found that defective lung branching was already apparent in *Wnt5a*$^{-/-}$ lungs at 11.5 dpc (Fig 1A–1C). The most striking feature of *Wnt5a*$^{-/-}$

lungs was the loss of proper position and orientation of the lung buds when the initial pattern was being generated. As lung development proceeded, the well-established programs that are dubbed domain branching and planar and orthogonal bifurcation were also impaired. The phenotype was completely penetrant.

In control lungs at 11.5 dpc, the 5 main branches designated as the right cranial (RCr), right middle (RMd), right caudal (RCd), right accessory (RAc), and left (L) branch were already fully separated (Fig 1A and 1S) [21]. Branching from these 5 branches would give rise to the 5 lobes (the cranial, middle, caudal, accessory lobes of the right lung, and a single left lobe of the left lung) in adult mice. RCr emerged at a more proximal position to that of RMd and RAc, which were at a similar axial level at this stage.

In *Wnt5a*-deficient lungs, the distance between the RCr and RMd branches was shortened due to the abnormal appearance of RCr at the axial level of RMd/RAc at 11.5 dpc (Fig 1B and 1C). To quantify the defects of RCr/RMd in *Wnt5a*$^{-/-}$ lungs, we measured the relative position of RCr and RMd (Fig 1J). We first determined the distance between RMd and RCr ($D_{RMd-RCr}$) and the distance between RMd and the bifurcation point (from the trachea) ($D_{RMd-bifurcation}$), respectively, and calculated their ratio ($R_{RCr-RMd}$). The relative distance ($R_{RCr-RMd}$) between RCr and RMd was reduced in *Wnt5a*$^{-/-}$ lungs.

The direction of the RMd and RAc branches relative to the RCd branch was also altered at 11.5 dpc. By contrast, the direction of RCr and left L1 (L.L1) was unaltered. We measured the angle between RMd and RCd ($\theta_{RMd-RCd}$) and the angle between RCd and RAc ($\theta_{RCd-RAc}$) (Fig 1K and 1L). In the absence of *Wnt5a*, $\theta_{RMd-RCd}$ was increased in 75% and reduced in 25% of the mutant lungs. $\theta_{RCd-RAc}$ was reduced in *Wnt5a*$^{-/-}$ lungs.

In approximately 25% of *Wnt5a*-deficient lungs, the distance between the left L1 (L.L1) and L2 (L.L2) branches was reduced at 11.5 dpc. Moreover, we observed a complex change in the position and direction of branches derived from RCr and L.L1 in *Wnt5a* mutant lungs. The founder branch for RCr and L.L1 were initially produced at the correct position and orientation and bifurcated to form the longitudinal and lateral branches. At 12.5 dpc, the daughter branches of RCr and L.L1 displayed defects in the position and direction where they branched. While the lateral branch from RCr and L.L1 ramified to form the main growth axis of the cranial and left lobes in control lungs, respectively, it was the longitudinal branch of RCr and L.L1 in *Wnt5a*$^{-/-}$ lungs that dominated the main growth axis of the corresponding lobe.

Daughter branches extended from the 5 main branches and subsequent branches also exhibited defective branching (Figs 1D–1F and 1G–1I and S1). At 12.5 dpc, the overall branching pattern of *Wnt5a*$^{-/-}$ lungs had diverged significantly from that in wild-type lungs. Together, these results suggest that *Wnt5a* signaling controls the position and direction of early lung branching.

We noticed a shortened trachea in *Wnt5a*$^{-/-}$ lungs and wondered whether reduced cell proliferation in lung epithelial cells could be related to the branching defects. Interestingly, loss of *Wnt5a* did not perturb proliferation of lung epithelial cells. No difference in EdU$^+$ epithelial cells between control and *Wnt5a*$^{-/-}$ lungs at 12.5 dpc was detected (Fig 1M–1R and 1T). This finding suggests that the primary defect in the absence of *Wnt5a* is likely changes in cellular organization.

## Loss of *Wnt5a* in either the lung mesenchyme or epithelium impairs branching morphogenesis

*Wnt5a* is expressed in both the lung epithelium and mesenchyme. To explore how *Wnt5a* in different niches controls lung branching, we selectively removed *Wnt5a* in the lung mesenchyme with the expectation that mesenchymal *Wnt5a* would trigger in part epithelial PCP

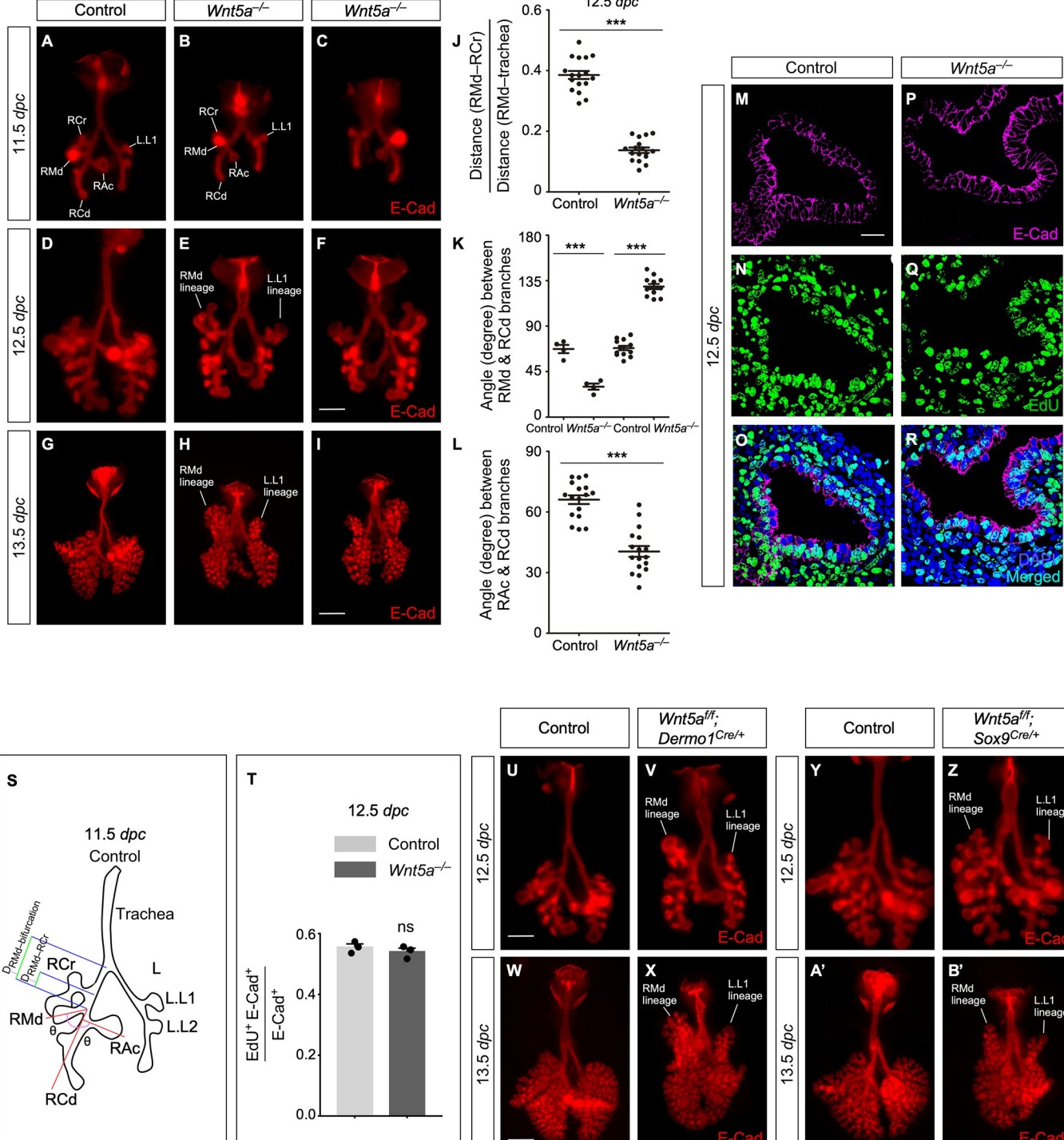

**Fig 1. Wnt5a controls the position and direction of lung branching.** (A-I) Ventral (A, B, D, E, G, H) and dorsal (C, F, I) views of dissected lungs from control and *Wnt5a*^−/−^ mouse embryos at the developmental stages indicated. E-Cad marked epithelial cells. (J) Quantification of the ratio of the distance of RMd–RCr to the distance of RMd–bifurcation (mean value ± SEM, unpaired Student's *t*-test, *n* = 17 pairs). (K) Quantification of the angle (in degrees) between RMd and RCd branches (mean value ± SEM, unpaired Student's *t*-test, *n* = 17 pairs). (L) Quantification of the angle (in degrees) between RAc and RCd branches (mean value ± SEM, unpaired Student's *t*-test, *n* = 17 pairs). (M-R) Immunostaining of lung sections collected from control and *Wnt5a*^−/−^ at 12.5 dpc. (S) Schematic diagram of the position and direction of lung branches in wild-type mice at 11.5 dpc. (T) Quantification of the cell proliferation rate in the epithelium of control and *Wnt5a*^−/−^ lungs at 12.5 dpc (mean value ± SEM, unpaired Student's *t*-test, *n* = 3 pairs). The rate of epithelial cell proliferation was calculated as the ratio of the number of EdU^+^ epithelial cells (EdU^+^E-Cad^+^) to the number of epithelial cells (E-Cad^+^). (U-B') Ventral views of dissected lungs from control, *Wnt5a*^f/f^; *Sox9*^Cre/+^ and *Wnt5a*^f/f^; *Dermo1*^Cre/+^ embryos at the developmental stages indicated. (\*\*\*) *p* < 0.001; ns, not significant. The underlying data for Fig

1J, 1K, 1L and 1T and the exact *P* values can be found in S1 Data. (Scale bars: A-F, 0.5 mm; G-I, 1 mm; M-R, 25 μm; U, V, Y, Z, 0.5 mm; W, X, A', B', 1 mm.) dpc, days post coitus; RAc, right accessory; RCd, right caudal; RCr, right cranial; RMd, right middle.

signaling. We produced $Wnt5a^{f/f}; Dermo1^{Cre/+}$ mice in which $Wnt5a$ was specifically eliminated in the lung mesenchyme by *Dermo1-Cre* [22]. $Wnt5a^{f/f}; Dermo1^{Cre/+}$ mice died soon after birth. Their lungs appeared compact compared to wild-type controls at 18.5 dpc and postnatal (p) day 0. The phenotype was highly penetrant and $Wnt5a^{f/f}; Dermo1^{Cre/+}$ mice exhibited branching defects. To further test this idea, we inspected $Wnt5a^{f/f}; Dermo1^{Cre/+}$ lungs at different stages of lung development. We found that the early branching defects in $Wnt5a^{f/f}; Dermo1^{Cre/+}$ lungs (Fig 1U–1X) were similar to those in $Wnt5a^{-/-}$ lungs described above.

We also tested whether $Wnt5a$ functions in the lung epithelium to control branching. To this end, we produced $Wnt5a^{f/f}; Shh^{Cre/+}$ mice. Unexpectedly, these animals were fully viable and, besides a mild digit phenotype, could not be distinguished from their wild-type littermates. To exclude the possibility that *Shh-Cre* [23] was inefficient in deleting $Wnt5a$, we generated $Wnt5a^{f/f}; Sox9^{Cre/+}$ mice. While *Sox9-Cre* [24] is activated a few days later than *Shh-Cre*, we suspect that *Sox9-Cre* could be more effective than *Shh-Cre* in removing $Wnt5a$ in the lung epithelium. Branching in $Wnt5a^{f/f}; Sox9^{Cre/+}$ lungs appeared normal at 11.5 dpc. Approximately 40% of $Wnt5a^{f/f}; Sox9^{Cre/+}$ mice exhibited lung defects (Fig 1Y–1B') similar to but milder than those in $Wnt5a^{-/-}$ lungs at 12.5 dpc. However, the branching defects in most $Wnt5a^{f/f}; Sox9^{Cre/+}$ lungs were not apparent until 13.5 dpc, reflecting the onset of *Sox9-Cre* expression at or after 11.5 dpc. Approximately 70% of $Wnt5a^{f/f}; Sox9^{Cre/+}$ mice displayed lung branching defects at 13.5 dpc, which were restricted to the lineage branches from RCr, RMd, and L.L1. Defective branching in the RCr, RMd, and L.L1 lineages did not emerge from their daughter branches, but from the subsequent secondary or tertiary branches. As a result, axis extension was only partially affected and the growth axis of the lobes was preserved.

Cell proliferation was unaltered in $Wnt5a^{f/f}; Sox9^{Cre/+}$ lungs at 13.5 dpc and $Wnt5a^{f/f}; Dermo1^{Cre/+}$ at 12.5 dpc (S2 Fig). Together, these results suggest that noncanonical *Wnt* signaling operates in both lung epithelium and mesenchyme to coordinate lung branching.

## Global elimination of *Vangl1/2* exhibits branching defects, similar to but milder than those due to loss of *Wnt5a*

To understand how the PCP pathway controls lung branching, we eliminated PCP signaling in the lung by generating mice deficient in both *Vangl1* and *Vangl2*. To this end, we set up crosses between $Vangl1^{gt/gt}; Vangl2^{f/+}; Sox2^{Cre/+}$ and $Vangl1^{gt/gt}; Vangl2^{f/f}$ mice and collected embryos at different developmental stages (10.5 to 18.5 dpc). $Vangl1^{gt}$ is a gene-trapped allele [25] that leads to a complete loss of *Vangl1* activity while a floxed (f) allele of *Vangl2* ($Vangl2^{f}$) [26] is converted into a null allele ($Vangl2^{-}$) upon Cre expression. We focused on $Vangl1^{gt/gt}; Vangl2^{f/f}; Sox2^{Cre/+}$ embryos (denoted as $Vangl1^{gt/gt}; Vangl2^{-/-}$ in this study) that are deficient in both *Vangl1* and *Vangl2*. Early ubiquitous expression of *Sox2-Cre* resulted in the production of $Vangl2^{-}$ from $Vangl2^{f}$ in all tissues (Fig 2A–2F). Of note, $Vangl1^{gt/gt}$ mice are viable and fertile without apparent phenotypes. We noticed that neither $Vangl2^{-/-}$ nor $Vangl1^{gt/+}; Vangl2^{-/-}$ lungs displayed branching defects despite the fact that all of these embryos had an open neural tube. This is consistent with a functional redundancy between *Vangl1* and *Vangl2* during lung branching. Such a dose-dependent effect of *Vangl1/2* levels on PCP signaling has been documented in several other tissues.

$Vangl1^{gt/gt}; Vangl2^{-/-}$ embryos exhibited an open neural tube and a shortened axis and died shortly after birth as stated in prior publications [26]. At 18.5 dpc, their lungs appeared more

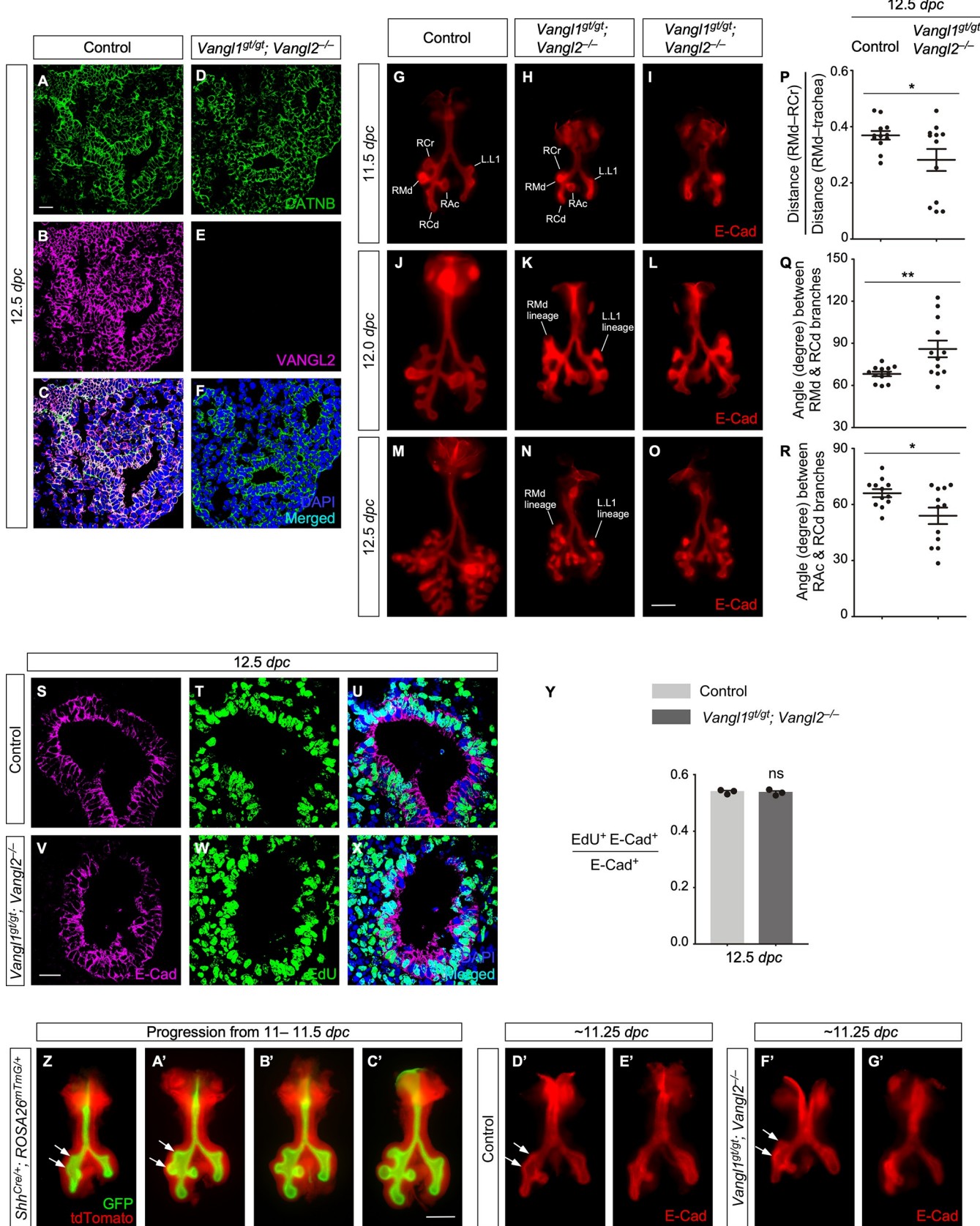

**Fig 2.** *Vangl1/2* **control the position and direction of lung branching.** (A-F) Immunostaining of lung sections collected from control and *Vangl1^{gt/gt}*; *Vangl2^{−/−}* mice at 12.5 dpc. CATNB (CTNNB1) labeled epithelial and mesenchymal cells. (G-O) Ventral (G, H, J, K, M, N) and dorsal (I, L, O) views of dissected lungs from control and *Vangl1^{gt/gt}*; *Vangl2^{−/−}* embryos at the developmental stages indicated. (P) Quantification of the ratio of the distance of RMd–RCr to the distance of RMd–bifurcation (mean value ± SEM, unpaired Student's *t*-test, *n* = 12 pairs). (Q) Quantification of the angle (in degrees) between RMd and RCd branches (mean value ± SEM, unpaired Student's *t*-test, *n* = 12 pairs). (R) Quantification of the angle (in degrees) between RAc and RCd branches (mean value ± SEM, unpaired Student's *t*-test, *n* = 12 pairs). (S-X) Immunostaining of lung sections collected from control and *Vangl1^{gt/gt}*; *Vangl2^{−/−}* mice at 12.5 dpc. (Y) Quantification of the cell proliferation rate in the epithelium of control and *Vangl1^{gt/gt}*; *Vangl2^{−/−}* lungs at 12.5 dpc (mean value ± SEM, unpaired Student's *t*-test, *n* = 3 pairs). The rate of epithelial cell proliferation was calculated as the ratio of the number of EdU^+ epithelial cells (EdU^+E-Cad^+) to the number of epithelial cells (E-Cad^+). (Z-C') Ventral views of dissected lungs from *Shh^{Cre/+}*; *ROSA26^{mTmG/+}* embryos at the developmental stages indicated. Note that A'-C' came from embryos within the same litter. Arrows point to enlarged lumen where RCr and RMd will emerge. (D'-G') Ventral views of dissected lungs from control and *Vangl1^{gt/gt}*; *Vangl2^{−/−}* embryos within the same litter at approximately 11.25 dpc. Arrows point to changes in the overall shape and direction of the epithelial sheet where RCr and RMd will emerge in *Vangl1^{gt/gt}*; *Vangl2^{−/−}* lungs. (*) *p* < 0.05; (**) *p* < 0.01; ns, not significant. The underlying data for Fig 2P, 2Q, 2R and 2Y and the exact *P* values can be found in S1 Data. (Scale bars: A-F, 25 μm; G-O, 0.5 mm; S-X, 25 μm; Z-G', 0.5 mm) dpc, days post coitus; RAc, right accessory; RCd, right caudal; RCr, right cranial; RMd, right middle.

compact than control lungs while the lumen diameter in the airways was reduced. This suggested defects in lung branching due to loss of PCP signaling.

We examined lung branching in *Vangl1^{gt/gt}*; *Vangl2^{−/−}* lungs at earlier stages. The first sign of defective branching detected at 11.5 dpc was misplacement of lung buds (Fig 2G–2I), similar to those observed in *Wnt5a^{−/−}* or *Wnt5a^{f/f}*; *Dermo1^{Cre/+}* lungs. However, subsequent branching defects in *Vangl1^{gt/gt}*; *Vangl2^{−/−}* lungs were not as pronounced (Figs 2J–2R and S3) as those in *Wnt5a^{−/−}* or *Wnt5a^{f/f}*; *Dermo1^{Cre/+}* lungs. These results suggest that *Vangl1/2* mediate *Wnt5a* signaling in controlling branching but *Wnt5a* has additional targets other than *Vangl1/ 2*. No difference in EdU^+ epithelial cells between control and *Vangl1^{gt/gt}*; *Vangl2^{−/−}* lungs at 12.5 dpc was detected (Fig 2S–2Y), again supporting a primary defect in cellular organization. No apparent defects in smooth muscle cells or blood vessels were detected in *Vangl1^{gt/gt}*; *Vangl2^{−/−}* lungs (S4 Fig).

To further understand how *Wnt5a–Vangl1/2* signaling controls lung branching, we traced lung development in control and *Vangl1/2* mutant lungs from 11.0 to 11.5 dpc. We found that the epithelium where RCr/RMd and L.L1/L.L2 emerge underwent coordinated morphological changes in control lungs (Fig 2Z–2C'). The lumen was enlarged first. Rudiments of RCr/RMd and L.L1/L.L2 were then formed. Meanwhile, the mesenchyme appeared to "push down" the epithelium between the two future branches. Finally, RCr/RMd and L.L1/L.L2 emerged at the defined position and direction. We speculate that the mechanical force between cells is affected in the absence of *Wnt5a–Vangl1/2* signaling. This could alter the overall shape and orientation of the epithelial sheet. As a result, the relative position and direction of RCr/RMd and L.L1/L. L2 were affected (Fig 2D'–2G').

Unlike *Wnt5a*, no apparent branching defects were observed in *Vangl1^{gt/gt}*; *Vangl2^{f/f}*; *Sox9^{Cre/+}* or *Vangl1^{gt/gt}*; *Vangl2^{f/f}*; *Dermo1^{Cre/+}* lungs (S5N–S5Q Fig). Although *Sox9-Cre* was expressed after early branching had initiated, epithelial *Vangl2* was efficiently removed in *Vangl2^{f/f}*; *Sox9^{Cre/+}* lungs by 14.5 dpc (S5A–S5M Fig) when active branching was proceeding. This suggests that coordination of *Vangl1/2* signaling (hence the downstream effectors) in both the epithelium and mesenchyme contribute to lung branching.

### *Foxa2* participates in transducing the *Wnt5a* signal during lung branching

The discrepancy in phenotypes between *Wnt5a* and *Vangl1/2* mutant lungs prompted us to search for *Wnt5a* targets other than *Vangl1/2*. We performed qPCR analysis on control and *Wnt5a*-deficient lungs to identify these potential targets. We found that *Foxa2* expression in the lung was significantly reduced in the absence of *Wnt5a* at 12.5 dpc (S6 Fig).

We then tested if *Wnt5a* regulated *Foxa2* expression through noncanonical or canonical pathways. *Foxa2* expression was unaltered in *Vangl1^{gt/gt}*; *Vangl2^{−/−}* lungs by qPCR analysis (S6

Fig), suggesting that *Foxa2* expression is not controlled by noncanonical *Wnt5a* signaling. Moreover, expression of β-catenin-related genes (such as *Axin2* and *Lef1*) was reduced in *Wnt5a*$^{-/-}$ lungs (S6 Fig). These results support the notion that *Wnt5a* controls *Foxa2* expression through the canonical pathway. Loss of *Foxa1* and *Foxa2* transcription factors has been reported to result in defective branching [27]. We speculate that *Wnt5a* coordinates lung branching by signaling through *Vangl1/2*, *Foxa2*, and other targets. The role of *Foxa2* in mediating *Wnt5a* function in lung branching requires future investigation.

### RNA-Seq and proteomic analysis reveals pathways that regulate focal adhesions, ECM–receptor interactions, and the actomyosin cytoskeleton during PCP-mediated lung branching

To uncover the molecular basis that underlies the branching defects in PCP mutant lungs, we performed RNA-Seq analysis of control, *Wnt5a*$^{f/f}$; *Dermo1*$^{Cre/+}$, *Wnt5a*$^{f/f}$; *Sox9*$^{Cre/+}$, and *Vangl1*$^{gt/gt}$; *Vangl2*$^{-/-}$ lungs. Pathway analysis revealed perturbed pathways in the mutant lungs that regulate focal adhesions, extracellular matrix (ECM)–receptor interactions and the actomyosin cytoskeleton (Fig 3A–3C). These pathways were also found to be perturbed through proteomic analysis of control and *Vangl1/2* knockout cells by mass spectrometry (Fig 3D). Control and *Vangl1/2* knockout cells were derived from control and *Vangl1*$^{gt/gt}$; *Vangl2*$^{-/-}$ embryonic lungs, respectively.

To further test the functional role of focal adhesions in lung branching, we examined expression of focal adhesion kinase (FAK) and phosphorylated FAK (p-FAK) [28] in control and *Wnt5a*- or *Vangl1/2*-deficient lungs (Figs 3E–3W and S7 and S8). p-FAK is an indicator of the activity of focal adhesion, a mechanical link between the ECM and intracellular actin bundles. FAK and p-FAK were widely expressed in lung epithelial cells, and the signal was concentrated along the apical and basal surface (Fig 3M, 3N, 3S and 3T). VANGL2 is expressed in all lung cells and is concentrated at the apical region of epithelial cells, where VANGL2 expression colocalizes with p-FAK and the actin cytoskeleton (S5A–S5M Fig). FAK expression and distribution were unaltered in the absence of PCP signaling (Fig 3P and 3Q). By contrast, the levels of p-FAK were significantly reduced albeit the subcellular distribution of p-FAK was unaltered in the lung epithelium of mutant lungs (Fig 3V and 3W). This conclusion was confirmed by Western blot analysis of lysates from *Wnt5a*- or *Vangl1/2*-deficient lungs (S9 Fig). The protein levels of p-FAK were significantly diminished in the mutant lungs and cells. These results suggest that FAK phosphorylation after cell activation is compromised in the absence of PCP signaling. They also suggest that cell-matrix adhesions (focal adhesions) are perturbed in *Wnt5a* and *Vangl1/2* knockout lungs. Without proper focal adhesions, lung epithelial cells would fail to undergo morphogenetic changes required to produce branches at the correct position and direction.

To better visualize focal adhesions, ECM and the actin cytoskeleton, we examined the cellular properties of control and *Vangl1/2* knockout cells on fibronectin-coated dishes. After culturing for 48 h, control cells already formed a well-organized network of actin cytoskeleton, which bound to the fibronectin through FAK (Fig 3X, 3Z and 3B'). p-FAK indicated successful cell activation through interactions with fibronectin. By contrast, these characteristic features were not detected in *Vangl1/2* knockout cells (Figs 3Y, 3A' and 3B' and **S9**). Introduction of the wild-type but not the mutant form of VANGL2 (VANGL2–84A) to *Vangl1/2* knockout cells restored p-FAK expression (Figs 3C' and **S9**). WNT5A induces phosphorylation of VANGL2 at position 84 and VANGL2–84A blocks signal transduction of WNT5A [29]. These results support a model in which a signaling cascade of *Wnt5a*–*Vangl1/2* controls focal adhesions during specification of the position and direction of lung branches.

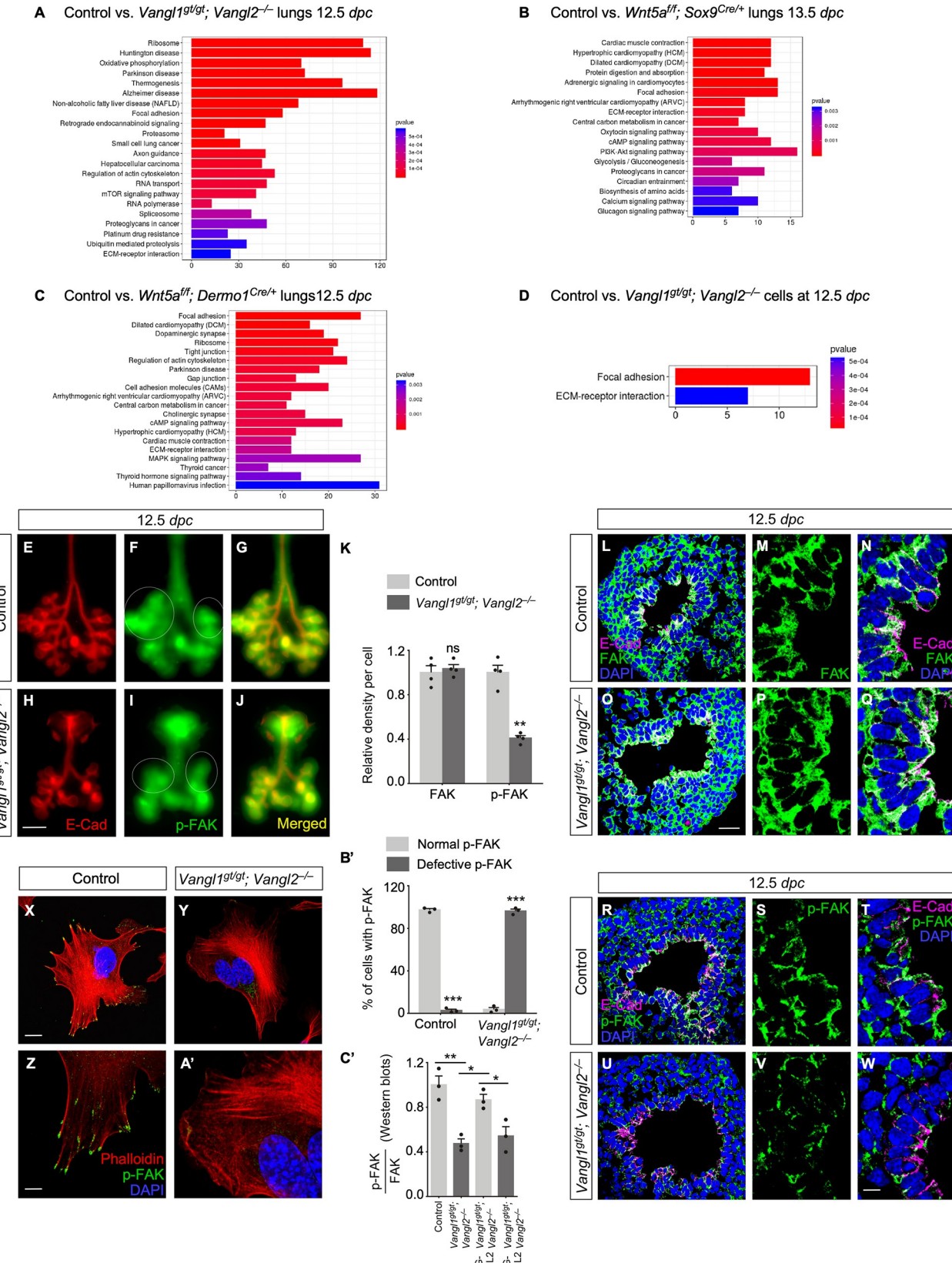

A  Control vs. *Vangl1gt/gt; Vangl2−/−* lungs 12.5 *dpc*

B  Control vs. *Wnt5af/f; Sox9Cre/+* lungs 13.5 *dpc*

C  Control vs. *Wnt5af/f; Dermo1Cre/+* lungs 12.5 *dpc*

D  Control vs. *Vangl1gt/gt; Vangl2−/−* cells at 12.5 *dpc*

**Fig 3. PCP signaling controls focal adhesions.** (A) KEGG pathway analysis of bulk RNA-Seq of control and *Vangl1^gt/gt^; Vangl2^−/−^* lungs at 12.5 dpc. Bulk RNA-Seq data were deposited to Gene Expression Omnibus database (GSE188993). (B) KEGG pathway analysis of bulk RNA-Seq of control and *Wnt5a^f/f^; Sox9^Cre/+^* lungs at 13.5 dpc. Bulk RNA-Seq data were deposited to Gene Expression Omnibus database (GSE188993). (C) KEGG pathway analysis of bulk RNA-Seq of control and *Wnt5a^f/f^; Dermo1^Cre/+^* lungs at 12.5 dpc. Bulk RNA-Seq data were deposited to Gene Expression Omnibus database (GSE188993). (D) KEGG pathway analysis of the proteomes of control and *Vangl1^gt/gt^; Vangl2^−/−^* cells. All mass spectrometry data can be found in the S1 Table and also are publicly available via the UCSD Mass Spectrometry Interactive Virtual Environment (MassIVE), a full member of the Proteome Exchange consortium, under the dataset number MSV000089851. (E-J) Ventral views of dissected lungs from control and *Vangl1^gt/gt^; Vangl2^−/−^* embryos at the developmental stages indicated. E-Cad marked epithelial cells. (K) Quantification of FAK and p-FAK signal per cell in control and *Vangl1^gt/gt^; Vangl2^−/−^* lungs at 12.5 dpc (mean value ± SEM, unpaired Student's *t*-test, *n* = 4 pairs). (L-W) Immunostaining of lung sections collected from control and *Vangl1^gt/gt^; Vangl2^−/−^* mice at 12.5 dpc. (X-A') Immunofluorescence of control and *Vangl1^gt/gt^; Vangl2^−/−^* cells. Phalloidin stained F-actin. (B') Quantification of normal and defective p-FAK in control and *Vangl1^gt/gt^; Vangl2^−/−^* cells (mean value ± SEM, unpaired Student's *t*-test, *n* = 3 pairs). (C') Quantification of the ratio of p-FAK to FAK in control cells, *Vangl1^gt/gt^; Vangl2^−/−^* cells, and *Vangl1^gt/gt^; Vangl2^−/−^* cells expressing VANGL2 or VANGL2 (84A) (mean value ± SEM, one-way ANOVA, *n* = 3 pairs). (*) $p < 0.05$; (**) $p < 0.01$; (***) $p < 0.001$; ns, not significant. The underlying data for Fig 3K, 3B' and 3C' and the exact *P* values can be found in S1 Data. (Scale bars: E-J, 0.5 mm; L, O, R, U, 25 μm; M, N, P, Q, S, T, V, W, 5 μm; X, Y, 10 μm; Z, A', 5 μm.) dpc, days post coitus; FAK, focal adhesion kinase; p-FAK, phosphorylated FAK.

We noted increased levels of phosphorylated Cofilin (p-Cofilin) [30] in *Vangl1/2* knockout lungs compared to controls (S10 Fig). This suggests that the assembly and disassembly of actin filaments regulated by Cofilin are affected in the mutant lungs. By contrast, the expression levels of Laminin, phosphorylated myosin light chain (pMLC) or F-actin [31] visualized by phalloidin were not significantly altered in the absence of *Vangl1/2* (S11 Fig).

## Perturbed focal adhesions in lung explants are associated with defective branching

To assess the functional role of focal adhesions during lung branching, we applied FAK inhibitor, PF-573228 [32], to lung explants to disrupt the function of focal adhesions (S12 Fig). After incubation with PF-573228 for 4 h, the branching direction and position of the middle lobe of lung explants was altered (Fig 4A–4L and 4Y). If lung explants were treated for 8 h, branching of most lobes was disrupted (Fig 4M–4R and 4Y). For instance, the position and direction of branches in the cranial lobe, middle lobe, and L.L1 were perturbed (Fig 4Y). E-Cad levels were reduced upon FAK inhibition (Fig 4S–4X). Together, these findings suggest that p-FAK (hence focal adhesions) is one of the many downstream effectors of *Wnt5a–Vangl1/2* signaling that contribute to lung branching.

## Discussion

Our studies have identified a *Wnt5a–Vangl1/2* axis as a key component that controls the position and direction of lung branching. In this model, cytoskeletal reorganization and changes in focal adhesions induced by *Wnt5a* signaling drive branching morphogenesis by establishing the position and direction of lung branches. These findings provide new mechanistic insights into lung branching morphogenesis (Fig 4Z). They will also serve as a paradigm for understanding the molecular pathways that regulate branching morphogenesis in other organs.

Our results support a model in which *Wnt5a* controls the position and direction of branching through *Vangl1/2*. However, the *Wnt5a–Vangl1/2* axis likely functions in a signaling network. In this scenario, *Wnt5a* has additional targets and *Foxa2* is not only regulated by other pathways but *Foxa2* also controls other processes. We propose that *Foxa2* is regulated by pathways other than *Wnt5a* signaling. In this regard, it is interesting to note that loss of both *Foxa1* and *Foxa2* disrupts branching morphogenesis, and epithelial cell proliferation and differentiation were inhibited [27]. Additional analysis is necessary to reveal the signaling network in which *Wnt5a* and *Vangl1/2* function.

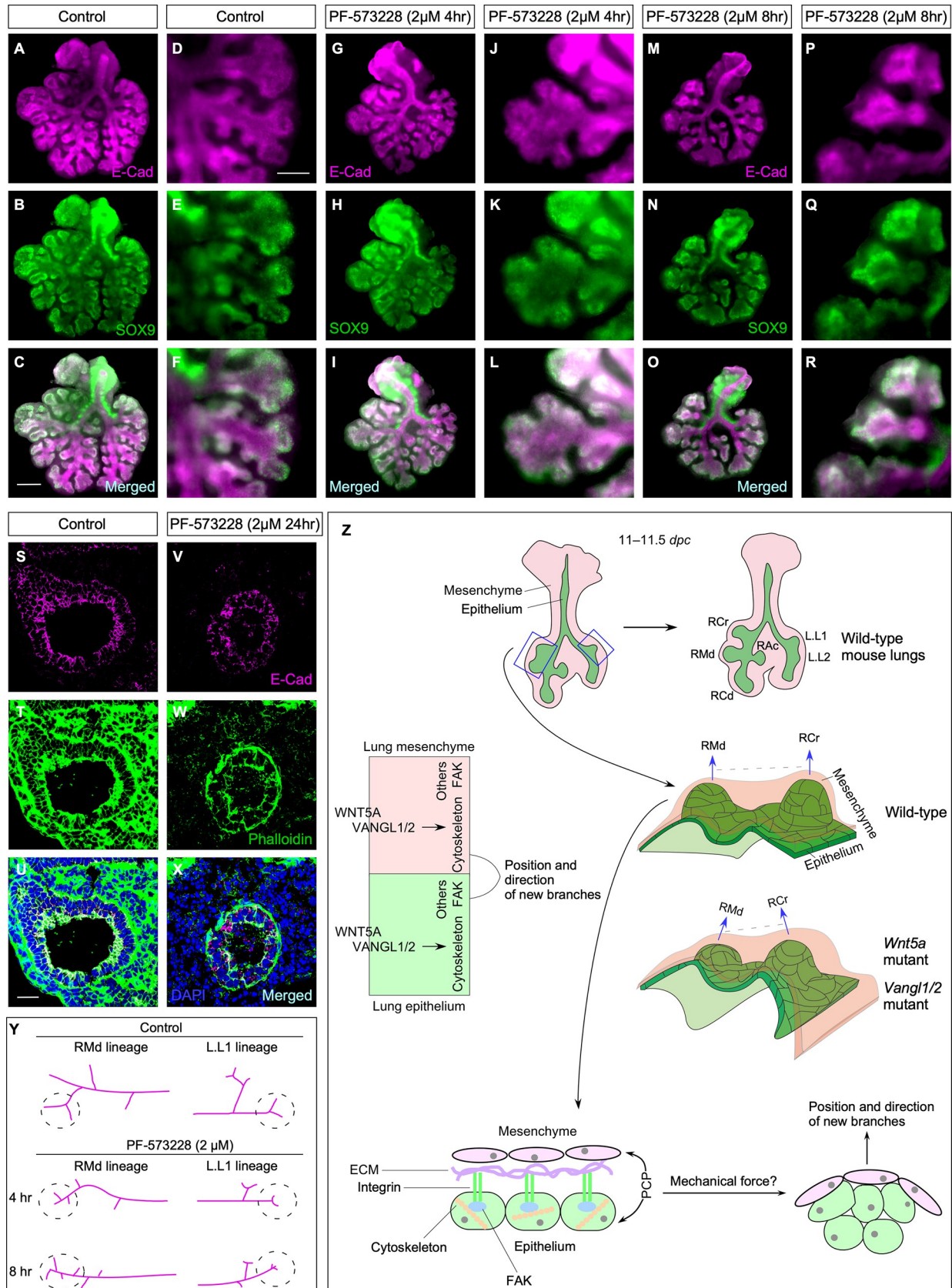

**Fig 4. Perturbation of focal adhesions in lung explants lead to branching defects.** (A-R) Immunostaining of lung explants treated with control media or PF-573228 (FAK inhibitor). E-Cad marked epithelial cells. (S-X) Immunostaining of lung explants treated with control media or PF-

573228. Phalloidin stained F-actin. (Y) Schematic diagram detailing the branching defects in lung explants treated with PF-573228. (Z) A model of how *Wnt5a–Vangl1/2* signaling controls the cytoskeleton and focal adhesions to specify the position and direction of lung branches. PCP signaling influences the cytoskeleton/FAK/integrin/ECM complex and induces mechanical force through the coordination between the epithelium and mesenchyme. This results in reorganization of the epithelial sheet (shape and orientation) and determines the position and direction of a new branch. Consistent with this model, loss of *Wnt5a* or *Vangl1/2* leads to incorrect positions and directions of RCr and RMd. (Scale bars: A-C, G-I, and M-O, 0.5 mm; D-F, J-L, and P-R, 0.125 mm; S-X, 25 μm.) dpc, days post coitus; ECM, extracellular matrix; FAK, focal adhesion kinase; L, left branch; PCP, planar cell polarity; RAc, right accessory; RCd, right caudal; RCr, right cranial; RMd, right middle.

We were somewhat surprised that the branching defects in *Vangl1^{gt/gt}; Vangl2^{−/−}* lungs were less severe than those in *Wnt5a^{−/−}* lungs. This finding suggests that PCP signaling is one of the several pathways employed during early lung branching to orchestrate the position and direction of lung branches. Likewise, it would be important to identify the cellular processes controlled by additional targets of *Wnt5a*.

The *Wnt5a–Vangl1/2* axis is an essential component of the machinery involved in selecting branch points and directing branch angles. Our results suggest that the actomyosin cytoskeleton and focal adhesions are perturbed in the absence of PCP signaling. We do not have the cellular resolution to reveal how these changes modify the collective behavior of cells near the prospective branch point to form a new branch. We speculate that PCP signaling influences the cytoskeleton/FAK/integrin/ECM complex and induces mechanical force [33]. Force production through the coordination between the epithelium and mesenchyme controls the shape and orientation of the epithelial sheet and consequently determines the position and direction of a new branch (Fig 4Z). To test this model would rely on further investigation into how coordination between the lung epithelium and mesenchyme leads to mechanical force production at the cellular level. However, other cellular processes mediated by PCP signaling could also contribute to the selection of branch position and direction. Additional genetic studies are also required to identify other pathways that interact with *Wnt5a–Vangl1/2* signaling to control branch position and direction. These investigations would provide insight into how cells at different locations respond differentially to *Wnt5a–Vangl1/2* signaling to initiate new lung branches. In this regard, it is interesting to note that certain lung branches are preferentially affected by *Wnt5a–Vangl1/2* signaling.

*Wnt5a* is expressed in multiple cell types in the developing lung as observed from in situ hybridization and single-cell RNA-seq. In this study, we have functionally defined the role of epithelial and mesenchymal *Wnt5a* in directing lung branching. However, the molecular and cellular events downstream of epithelial and mesenchymal *Wnt5a* signaling are unclear. Moreover, whether *Wnt5a* in other cell types, such as endothelial cells and pericytes, also regulates lung branching requires additional studies using genetic and molecular approaches. It is anticipated that the *Wnt5a* signal is received by a distinct subset of receptors/coreceptors and participates in a different signaling network in a given tissue or biological process.

Both *Vangl1^{gt/gt}; Vangl2^{f/f}; Shh^{Cre/+}* and *Vangl1^{gt/gt}; Vangl2^{f/f}; Sox9^{Cre/+}* mice exhibit no branching defects although *Vangl1^{gt/gt}; Vangl2^{f/f}; Sox9^{Cre/+}* mice subsequently develop alveolar phenotypes [34]. *Shh-Cre* fails to efficiently remove *Vangl2* while the late onset of *Sox9-Cre* expression is incapable of eliminating *Vangl2* before early branching ensues. Previous studies failed to uncover phenotypes in the trachea of *Wnt5a^{f/f}; Shh^{Cre/+}* mice [35]. This is consistent with our findings in which no defects in tracheal development or lung branching were observed. This was due to inefficient removal of *Wnt5a* by *Shh-Cre* and again highlights the importance of employing multiple Cre lines for conditional inactivation [34,36].

In summary, our work in this study has addressed the fundamental question of how lung branches are produced in the defined three-dimensional space. Identification of the signaling network coupled with genetic and cell biological studies will yield additional insight into this important issue.

## Materials and methods

### Animal husbandry

All the mouse experiments were performed following the protocols approved by the Institutional Animal Care and Use Committee (IACUC) of the University of California, San Francisco (UCSF) (AN187712). The mouse genotypes and ages were indicated in the main text and figures. The following mouse lines were used in this study: *Vangl1*$^{gt}$ [*Vangl1*$^{GT(XL802)Byg}$] and *Vangl2*$^{f}$ [*Vangl2*$^{tm1.1Yy}$] were provided by Dr. Yingzi Yang [26]. *Wnt5a*$^{f}$ [*B6;129S-Wnt5a*$^{tm1.1Krvl}$/J], *ROSA26*$^{mTmG}$ [*Gt(ROSA)26Sor*$^{tm4(ACTB-tdTomato,-EGFP)Luo}$/J], *Sox2-Cre* [*B6.Cg-Edil3*$^{Tg(Sox2-cre)1Amc}$/J] and *Shh-Cre* [*B6.Cg-Shh*$^{tm1(EGFP/cre)Cjt}$/J] were obtained from Jackson Laboratory (Bar Harbor, ME, USA). *Dermo1-Cre* [*Twist2*$^{tm1.1(cre)Dor}$/J] was obtained from Dr. David Ornitz [22]. *Sox9-Cre* [*Sox9*$^{tm3(Cre)Crm}$] was obtained from Dr. Benoit de Crombrugghe [24].

### Immunohistochemistry

Immunofluorescence was performed as previously described [34,37,38]. In brief, mouse embryonic lungs were collected at the indicated time points and fixed with 4% paraformaldehyde (PFA) on ice for 1 h. The samples were embedded in OCT and sectioned at 7 μm. The primary antibodies used were as follows: rat anti-E cadherin (1:200, Life Technologies, Cat# 13–1900, RRID:AB_86571), mouse anti-β-catenin (1:100, BD Transduction Laboratories, Cat# 610154, RRID:AB_397555), rat anti-PECAM1(MEC13.3) (1:100, Santa Cruz Biotechnology, Cat# sc-18916, RRID:AB_627028), rat anti-VANGL2 (1:100, MilliporeSigma, Cat# MABN750, RRID:AB_2721170), rabbit anti-FAK (1:100, Cell Signaling Technology, Cat# 3285S, RRID: AB_2269034), rabbit anti-Phospho-FAK (Tyr397) (1:100, Cell Signaling Technology, Cat# 3283S, RRID:AB_2173659), rabbit anti-Cofilin (1:150, Cell Signaling Technology, Cat# 5175S, RRID:AB_10622000), rabbit anti-phospho-Cofilin (Ser3) (1:100, Cell Signaling Technology, Cat# 3313S, RRID:AB_2080597), rabbit anti-Laminin (1:150, Sigma-Aldrich, Cat# L9393, RRID:AB_477163), rabbit anti-pMLC (S19) (1:100, Cell Signaling Technology, Cat# 3671S, RRID:AB_330248), mouse anti-ACTA2 (1:200, Thermo Scientific Lab Vision, Cat# MS-113-P0, RRID:AB_64001), and chicken anti-GFP (1:200, abcam, Cat# ab13970, RRID: AB_300798). Secondary antibodies and conjugates used were as follows: donkey anti-rabbit Alexa Fluor 488 or 594 (1:1,000, Life Technologies), donkey anti-mouse Alexa Fluor 488 or 594 (1:1,000, Life Technologies), and donkey anti-rat Alexa Fluor 594 (1:1,000, Life Technologies). The biotinylated secondary antibodies used were goat anti-hamster (1:1,000, Jackson ImmunoResearch Laboratories), donkey anti-rabbit (1:1,000, Jackson ImmunoResearch Laboratories), donkey anti-rat (1:1,000, Jackson ImmunoResearch Laboratories), and horse anti-mouse (1:1,000, Jackson ImmunoResearch Laboratories). The signal was detected using streptavidin-conjugated Alexa Fluor 488, 594, or 647 (1:1,000, Life Technologies) or HRP-conjugated streptavidin (1:1,000, Perkin-Elmer) coupled with fluorogenic substrate Alexa Fluor 594 or 488 tyramide for 30 s (1:200, TSA kit; Perkin Elmer). F-actin was stained with rhodamine-conjugated phalloidin (1:200, MilliporeSigma) in PBS for 2 h.

Confocal images were captured using a Leica SPE laser-scanning confocal microscope. Adjustment of red/green/blue/gray histograms and channel merges were performed using LAS AF Lite software (Leica Microsystems).

### Whole mount immunostaining of mouse lungs

Whole mount immunostaining of embryonic lungs was performed as previously described [36]. Briefly, the whole embryonic lungs were dissected out and fixed in 4% PFA on ice for 1 h.

Lungs were washed with PBS and dehydrated in graded methanols (25%, 50%, 75%, 100%). After incubating in 5% $H_2O_2$/methanol for 4 h, the samples were then rehydrated through graded methanols (100%, 75%, 50%, 25%, 0%) diluted in 0.1% Tween-20/PBS and incubated with blocking buffer (1.5% BSA/0.5% Triton X-100/PBS) for 2 h. The samples were then incubated with primary antibodies at 4˚C overnight. The primary antibodies used were as follows: rat anti-E cadherin (1:200, Life Technologies, Cat# 13–1900), goat anti-SOX9 (1:100, R&D Systems, Cat# AF3075, RRID:AB_2194160), rabbit anti-FAK (1:100, Cell Signaling Technology, Cat# 3285S, RRID:AB_2269034), and rabbit anti-Phospho-FAK (Tyr397) (1:100, Cell Signaling Technology, Cat# 3283S, RRID:AB_2173659). On the second day, the samples were washed with blocking buffer for 5 h, then incubated with secondary antibodies at 4˚C overnight. The final day, the samples were washed for 5 h with blocking buffer. Images were captured using a Nikon Eclipse E1000 microscope with SPOT 2.3 CCD camera.

## Cell proliferation assays

The proliferation of embryonic lung cells was assessed by EdU incorporation as previously described [36]. Pregnant females at indicated time points were intraperitoneally injected with appropriate EdU/PBS solution for 1 h before collection. The Click-iT EdU Alexa Fluor 488 Imaging Kit (Life Technologies) was used to assess EdU incorporation. The sections were costained with the epithelial cell marker E-Cadherin (E-Cad). The proliferation rate was calculated as the ratio of (EdU$^+$E-Cad$^+$ cells)/(E-Cad$^+$ cells).

## RNA-seq analysis

RNA-seq was performed as previously described [34]. Briefly, the embryonic lungs from *Vangl1$^{gt/gt}$; Vangl2$^{-/-}$* mice at 12.5 dpc, *Wnt5a$^{f/f}$; Dermo1$^{Cre/+}$* mice at 12.5 dpc and *Wnt5a$^{f/f}$; Sox9$^{Cre/+}$* at 13.5 dpc were lysed in 0.5 ml TRIzol (Life Technologies) and 100 μl chloroform was then added. After centrifugation at 4˚C for 15 min, the upper aqueous layer was collected and mixed with an equal volume of 70% ethanol. RNA was extracted with the RNeasy Mini Kit (Qiagen) following the manufacturer's instructions. RNA quality was evaluated using an Agilent 2100 Bioanalyzer. Samples were sequenced on an Illumina HiSeq 2000 or HiSeq4000. Differential gene expression, gene ontology (GO) enrichment analyses, and the barplot of gene ontology enrichment were performed using RStudio. Datasets have been deposited in NCBI's Gene Expression Omnibus database and are accessible through GEO Series accession number (GSE188993).

## Mass spectrometry data acquisition and analysis

**Sample preparation.**   Cell pellets were thawed on ice and subjected to sample preparation with the PreOmics iST kit (PreOmics, Planegg, Germany) according to the manufacturer's protocol. Samples were resuspended in LC-Load Buffer from the iST kit and peptide concentration determined (Pierce Quantitative Colorimetric or Fluorescent Peptide Assay, Thermo-Fisher Scientific, Waltham, Massachusetts). Sample concentration was normalized to 100 ng/μl and 2 μl was loaded onto the instrument.

**Mass spectrometry analysis—Liquid chromatography and timsTOF Pro.**   A nanoElute was attached in line to a timsTOF Pro equipped with a CaptiveSpray Source (Bruker, Hamburg, Germany). Chromatography was conducted at 40˚C through a 25-cm reversed phase C18 column (PepSep) at a constant flow rate of 0.5 μl/min. Mobile phase A was 98/2/0.1% Water/MeCN/Formic Acid (v/v/v) and phase B was MeCN with 0.1% Formic Acid (v/v). During a 108-min method, peptides were separated by a 3-step linear gradient (5% to 30% B over 90 min, 30% to 35% B over 10 min, 35% to 95% B over 4 min) followed by a 4-min isocratic

flush at 95% for 4 min before washing and a return to low organic conditions. Experiments were run as data-dependent acquisitions with ion mobility activated in PASEF mode. MS and MS/MS spectra were collected with $m/z$ 100 to 1,700 and ions with $z = +1$ were excluded.

Raw data files were searched using PEAKS Online Xpro 1.6 (Bioinformatics Solutions, Waterloo, Ontario, Canada). The precursor mass error tolerance and fragment mass error tolerance were set to 20 PPM and 0.02, respectively. The trypsin/Lys-C digest mode was set to semispecific and missed cleavages were set to 2. The human Swiss-Prot reviewed (canonical) database (downloaded from UniProt) and the common repository of adventitious proteins (cRAP, downloaded from The Global Proteome Machine Organization) totaling 20,487 entries were used. Carbamidomethylation was selected as a fixed modification. Deamidation (NQ) and Oxidation (M) were selected as variable modifications. A maximum of 3 variable modifications were allowed.

All experiments were repeated in biological triplicate and technical duplicate and subjected to the following filtration criteria:

1. During the PEAKS Online Xpro export process, a false discovery rate (FDR) cutoff for peptide identification was applied, and only peptides with FDR ≤1% were included.

2. Proteins were groups were required to have a FDR ≤1%.

All mass spectrometry data can be found in the S1 Table and also are publicly available via the UCSD Mass Spectrometry Interactive Virtual Environment (MassIVE), a full member of the Proteome Exchange consortium, under the dataset number MSV000089851.

## qPCR analysis

The qPCR assay was performed as previously described [34]. RNAs were extracted with the RNeasy Mini Kit (Qiagen) following the manufacturer's instructions. The extracted RNAs were reverse-transcribed with the Maxima First Strand cDNA Synthesis Kit (Thermo Scientific). Quantitative PCR (qPCR) was performed on an Applied Biosystems QuantStudio 5 Real-Time PCR System. Primers for qPCR are as follows:

mouse *Foxa2* forward, 5′-CATGGGACCTCACCTGAGTC-3′; reverse, 5′-CATCGAGTT CATGTTGGCGTA-3′, mouse *Axin2* forward, 5′-ATGGAGTCCCTCCTTACCGCAT-3′; reverse, 5′-GTTCCACAGGCGTCATCTCCTT-3′, mouse *Lef1* forward, 5′-ACTGTCAGGC GACACTTCCATG-3′; reverse, 5′-GTGCTCCTGTTTGACCTGAGGT-3′, mouse *Mmp9* forward, 5′-GACATAGACGGCATCCAGTATC-3′; reverse, 5′-GGTATAGTGGGACACATAG TGG-3′, mouse *Bmp4* forward, 5′-CGAGCCAACACTGTGAGG-3′; reverse, 5′-GAAGAGGA AACGAAAAGCAGAG-3′, mouse *Vegfa* forward, 5′-GGCAAAGTGACTGACCTGCT-3′; reverse, 5′-CTGTCTGTCTGTCCGTCAGC-3′, mouse *Gapdh* forward, 5′-AGGTTGTCTCC TGCGACTTCA-3′; reverse, 5′-CCAGGAAATGAGCTTGACAAAGTT-3′.

## Derivation and culture of primary lung cells

Control and *Vangl1*$^{gt/gt}$; *Vangl2*$^{-/-}$ embryos at 12.5 dpc were dissected, and the lungs were digested in 0.1% trypsin–EDTA at 37˚C for 20 min. The cells were then incubated in DMEM containing 10% FBS, 1× penicillin/streptomycin and 1× L-glutamine.

## Lentiviral production and transduction

3xFLAG-Vangl2 and 3xFLAG-Vangl2-84A (a gift from Dr. Yingzi Yang) [29] were cloned into the modified pSECC lentiviral vector (the *eEF1a* promoter replaces the cassette of [gRNA site–FLAG-SV40NLS–Cas9–NLS–P2A–Cre]).

Lentiviruses were produced as previously described [34]. HEK293T cells were plated at 60% confluence in 10-cm dishes 24 h before transfection. For transfection, 2 μg of pMD2.G, 2 μg of psPAX2, and 5 μg of the lentiviral plasmid (3xFLAG-Vangl2 or 3xFLAG-Vangl2-84A) were mixed in 1000 μl OPTI-MEM with 50 μl of polyethylenimine (PEI) (1 mg/ml) and added to HEK293T cells when they reached 80% to 90% confluence. Approximately 48 h post-transfection, the viral supernatants were collected, filtered through 0.45-μm PVDF membrane filters, then added to control and *Vangl1$^{gt/gt}$; Vangl2$^{-/-}$*-adherent lung primary cells together with 8 μg/ml polybrene. Media were replaced 12 h post-transduction.

## Co-immunoprecipitation

The co-immunoprecipitation assay was performed as previously described [39]. *Vangl1/2* knockout cell lines that stably express 3xFLAG-Vangl2 or 3xFLAG-Vangl2-84A were seeded onto 10 cm dishes until they reached 100% confluence. Cells were lysed in immunoprecipitation buffer (1% Triton X-100, 150 mM NaCl, 50 mM Tris-Cl at pH 7.5, 1 mM EDTA, protease inhibitor cocktail). The samples were centrifuged at 12,000 rpm for 15 min at 4˚C. The supernatants were removed and bound to 20 μl of anti-FLAG M2 beads (Sigma) overnight at 4˚C with nutating. Beads were washed 3 times with immunoprecipitation buffer and eluted with SDS sample buffer, then analyzed by Western blotting.

## Western blotting analysis

Embryonic lung tissues were pipetted in RIPA buffer with 1× protease inhibitor cocktail and 1× PMSF. The lysates were centrifuged at 13,200 rpm at 4˚C for 15 min, then analyzed by Western blot as previously described [34]. The primary antibodies used were as follows: mouse anti-FLAG M2 (1:3,000, MilliporeSigma, Cat# F3165, RRID:AB_259529), rabbit anti-FAK (1:1,000, Cell Signaling Technology, Cat# 3285S, RRID:AB_2269034), rabbit anti-p-FAK (Tyr397) (1:1,000, Cell Signaling Technology, Cat# 3283S, RRID:AB_2173659), and mouse anti-alpha-tubulin (1:3,000, Developmental Studies Hybridoma Bank, Cat# 12G10, RRID: AB_1157911).

## Culture of mouse embryonic lungs

Culture of mouse embryonic lungs was performed as previously described [36]. Briefly, embryonic lungs were dissected from wild-type mice at 11.5 dpc and placed on top of the polycarbonate nuclepore membranes (Millipore), which were floating in cultured medium (DMEM/F-12 supplemented with penicillin/streptomycin, L-glutamine and 1% FBS) with or without the FAK inhibitor, PF-573228 (Selleck Chemicals) (2 μM in DMSO) for 4 h or 8 h. A 24-well plate that contained the samples was replaced with fresh media and then cultured for another 48 h. The lungs were then collected for RNA-Seq or for imaging.

## Statistical analysis

All the biological repeats we performed were more than or equal to 3, and the detailed biological replicates (n numbers) were indicated in the figure legends. All the statistical comparisons between different groups were shown as mean value ± SEM. Two-tailed Student's *t*-tests and one-way ANOVA were applied to calculate the *P* values and the statistical significance was evaluated as $^*$ $P < 0.05$, $^{**}$ $P < 0.01$, and $^{***}$ $P < 0.001$.

## Supporting information

**S1 Fig. Loss of *Wnt5a* leads to defective branching morphogenesis.** Ventral (A, B, D, E) and dorsal (C) views of dissected lungs from wild-type and *Wnt5a*$^{-/-}$ embryos at the developmental stages indicated. Ventral (F, G, I, J) and dorsal (H) views of dissected lungs from wild-type and *Wnt5a*$^{f/f}$*; Sox9*$^{Cre/+}$ embryos at the developmental stages indicated. Ventral (K, L, N, O) and dorsal (M) views of dissected lungs from wild-type and *Wnt5a*$^{f/f}$*; Dermo1*$^{Cre/+}$ embryos at the developmental stages indicated. (Scale bars: A-C, F-H, and K-M, 1 mm; D, E, I, J, N, and O, 1 mm.) dpc, days post coitus.
(PDF)

**S2 Fig. The rate of cell proliferation is unaltered in the absence of *Wnt5a*.** (A-F) Immunostaining of lung sections collected from control and *Wnt5a*$^{f/f}$*; Sox9*$^{Cre/+}$ mice at 13.5 dpc. (G) Quantification of the cell proliferation rate in the epithelium of control and *Wnt5a*$^{f/f}$*; Sox9*$^{Cre/+}$ lungs at 13.5 dpc (mean value ± SEM, unpaired Student's *t*-test, *n* = 3 pairs). The rate of epithelial cell proliferation was calculated as the ratio of the number of EdU$^+$ epithelial cells (EdU$^+$E-Cad$^+$) to the number of epithelial cells (E-Cad$^+$). (H-M) Immunostaining of lung sections collected from control and *Wnt5a*$^{f/f}$*; Dermo1*$^{Cre/+}$ mice at 12.5 dpc. (N) Quantification of the cell proliferation rate in the epithelium of control and *Wnt5a*$^{f/f}$*; Dermo1*$^{Cre/+}$ lungs at 12.5 dpc (mean value ± SEM, unpaired Student's *t*-test, *n* = 3 pairs). The underlying data for S2G and S2N Fig and the exact *P* values can be found in S1 Data. (Scale bars: A-F and H-M, 25 μm.) dpc, days post coitus; ns, not significant.
(PDF)

**S3 Fig. *Vangl1/2* control the position and direction of lung branching.** Ventral (A, B, D, E, G, H) and dorsal (C, F, I) views of dissected lungs from wild-type and *Vangl1*$^{gt/gt}$*; Vangl2*$^{-/-}$ embryos at the developmental stages indicated. (Scale bars: A-C, 0.5 mm; D-F, 1 mm; G-I, 1 mm.)
(PDF)

**S4 Fig. Smooth muscle cells and blood vessels are unaffected in *Vangl1/2* mutant lungs.** (A-D) Immunostaining of lung sections collected from control and *Vangl1*$^{gt/gt}$*; Vangl2*$^{-/-}$ mice at 15.5 dpc. SMA marks smooth muscle cells; CD31 labels endothelial cells. (Scale bar: A-D, 25 μm.) dpc, days post coitus.
(PDF)

**S5 Fig. Selective loss of *Vangl1/2* in the lung epithelium or mesenchyme does not lead to branching defects.** (A-F) Immunostaining of lung sections collected from control and *Wnt5a*$^{-/-}$ mice at 12.5 dpc. (G) Quantification of VANGL2 signal in lung cells of control or *Wnt5a* mutant lungs (mean value ± SEM, unpaired Student's *t*-test, *n* = 4 pairs). (H-M) Immunostaining of lung sections collected from control and *Vangl2*$^{f/f}$*; Sox9*$^{Cre/+}$ lungs at 14.5 dpc. (N-Q) Ventral views of dissected lungs from control and mutant lungs at 14.5 dpc. The underlying data for S5G Fig and the exact *P* value can be found in S1 Data. (Scale bar: A-F, H-M, 25 μm; N-Q, 1 mm) dpc, days post coitus; ns, not significant.
(PDF)

**S6 Fig. The transcription factor, *Foxa2*, and major signaling pathways in the developing lungs are perturbed in the absence of *Wnt5a*.** (A) qPCR analysis of *Foxa2* transcript levels in control and *Wnt5a*$^{-/-}$ lungs at 12.5 and 13.5 dpc (mean value ± SEM, unpaired Student's *t*-test, *n* = 3 pairs). (B) qPCR analysis of *Foxa2* transcript levels in control and *Vangl1*$^{gt/gt}$*; Vangl2*$^{-/-}$ lungs at 12.5 and 13.5 dpc (mean value ± SEM, unpaired Student's *t*-test, *n* = 3 pairs). (C) qPCR analysis of components of the major signaling pathways in control and

*Wnt5a*$^{-/-}$ lungs at 12.5 dpc (mean value ± SEM, unpaired Student's *t*-test, *n* = 3 pairs). (*)
*p* < 0.05; (**) *p* < 0.01. The underlying data for S5A–S5C Fig and the exact *P* values can be
found in S1 Data. dpc, days post coitus; ns, not significant.
(PDF)

**S7 Fig. p-FAK is reduced in *Wnt5a*-null lungs.** (A-L) Whole-mount immunostaining of dissected lungs from control and *Wnt5a*$^{-/-}$ mice at 12.5 dpc. Lung epithelium was visualized by
E-cadherin (E-Cad). Circles in (H, K) indicate defective branching in *Wnt5a*-deficient lungs.
(M-T) Immunostaining of lung sections collected from control and *Wnt5a*$^{-/-}$ mice at 12.5
dpc. (Scale bars: A-L, 0.5 mm; M-T, 25 μm.) dpc, days post coitus; p-FAK, phosphorylated
FAK.
(PDF)

**S8 Fig. p-FAK is reduced in *Wnt5a*-deficient lung compartments.** (A-L) Immunostaining of
lung sections collected from control and *Wnt5a*$^{f/f}$; *Sox9*$^{Cre/+}$ lungs at 13.5 dpc. Lung epithelium
was visualized by E-cadherin (E-Cad). (M-X) Immunostaining of lung sections collected from
control and *Wnt5a*$^{f/f}$; *Dermo1*$^{Cre/+}$ lungs at 12.5 dpc. (Scale bar: A-X, 25 μm.) dpc, days post
coitus; p-FAK, phosphorylated FAK.
(PDF)

**S9 Fig. p-FAK is reduced in the absence of *Wnt5a* or *Vangl1/2*.** (A) Western blot analysis of
cell lysates derived from control and *Wnt5a*$^{-/-}$ lungs at 12.5 dpc. (B) Western blot analysis of
cell lysates derived from control and *Vangl1*$^{gt/gt}$; *Vangl2*$^{-/-}$ lungs at 12.5 dpc. (C) Western blot
analysis of cell lysates derived from control cells, *Vangl1*$^{gt/gt}$; *Vangl2*$^{-/-}$ cells and *Vangl1*$^{gt/gt}$;
*Vangl2*$^{-/-}$ cells expressing VANGL2 or VANGL2 (84A) as indicated. α-tubulin serves as the
loading control. dpc, days post coitus; FAK, focal adhesion kinase; p-FAK, phosphorylated
FAK.
(PDF)

**S10 Fig. p-Cofilin levels are increased in the absence of *Vangl1/2*.** (A-L) Immunostaining of
lung sections collected from control and *Vangl1*$^{gt/gt}$; *Vangl2*$^{-/-}$ mice at 12.5 dpc. Lung epithelium was marked by E-cadherin (E-Cad). (M) Quantification of the relative density of Cofilin
and p-Cofilin in lung cells at the epithelial tip or in the mesenchyme (mean value ± SEM,
unpaired Student's *t*-test, *n* = 4 pairs). (**) *p* < 0.01. The underlying data for S10M Fig and the
exact *P* values can be found in S1 Data. (Scale bar: A-L, 25 μm.) dpc, days post coitus; ns, not
significant; p-Cofilin, phosphorylated Cofilin;
(PDF)

**S11 Fig. Expression level of Laminin, F-actin, pMLC are unaltered in *Vangl1/2* mutant
lungs.** (A-X) Immunostaining of lung sections collected from control and *Vangl1*$^{gt/gt}$;
*Vangl2*$^{-/-}$ mice at 12.5 dpc. Lung epithelium was marked by E-cadherin (E-Cad). F-actin
was labeled by phalloidin. (Y) Quantification of phalloidin signal in lung cells of control or
*Vangl1/2* mutant lungs (mean value ± SEM, unpaired Student's *t*-test, *n* = 4 pairs). (Z) Quantification of pMLC signal in lung cells of control or *Vangl1/2* mutant lungs (mean value ± SEM,
unpaired Student's *t*-test, *n* = 4 pairs). The underlying data for S11Y and S11Z Fig and the
exact *P* values can be found in S1 Data. (Scale bar: A-X, 25 μm.) dpc, days post coitus; ns, not
significant; pMLC, phosphorylated myosin light chain.
(PDF)

**S12 Fig. p-FAK is reduced in the presence of FAK inhibitor.** (A-X) Immunostaining of lung
explants treated with control media or media containing 0.5 μM or 2 μM of PF-573228 (FAK
inhibitor) as indicated. (Y) Quantification of the relative density of FAK and p-FAK in lung

cells treated with PF-573228 (mean value ± SEM, one-way ANOVA, $n = 5$ pairs). (*) $p < 0.05$; (**) $p < 0.01$; (***) $p < 0.001$. The underlying data for S12Y Fig can be found in S1 Data. (Scale bar: A-X, 25 μm.) FAK, focal adhesion kinase; ns, not significant; p-FAK, phosphorylated FAK.
(PDF)

**S1 Raw Images. Raw images of Western blots.**
(PDF)

**S1 Data. Numerical data.**
(XLSX)

**S1 Table. Mass spectrometry data.**
(XLSX)

## Acknowledgments

Some data for this study were acquired at the Nikon Imaging Center at CVRI.

## Author Contributions

**Conceptualization:** Kuan Zhang, Erica Yao, Pao-Tien Chuang.

**Data curation:** Kuan Zhang, Erica Yao, Ethan Chuang, Biao Chen, Evelyn Y. Chuang, Regan F. Volk, Katherine L. Hofmann, Balyn Zaro, Pao-Tien Chuang.

**Formal analysis:** Kuan Zhang, Erica Yao, Ethan Chuang, Biao Chen, Regan F. Volk, Katherine L. Hofmann, Balyn Zaro, Pao-Tien Chuang.

**Funding acquisition:** Pao-Tien Chuang.

**Investigation:** Kuan Zhang, Erica Yao, Pao-Tien Chuang.

**Methodology:** Pao-Tien Chuang.

**Project administration:** Pao-Tien Chuang.

**Resources:** Pao-Tien Chuang.

**Supervision:** Pao-Tien Chuang.

**Validation:** Kuan Zhang, Erica Yao, Ethan Chuang, Biao Chen, Evelyn Y. Chuang, Regan F. Volk, Katherine L. Hofmann, Balyn Zaro, Pao-Tien Chuang.

**Visualization:** Kuan Zhang, Erica Yao, Ethan Chuang, Evelyn Y. Chuang, Pao-Tien Chuang.

**Writing – original draft:** Kuan Zhang, Erica Yao, Pao-Tien Chuang.

**Writing – review & editing:** Kuan Zhang, Erica Yao, Ethan Chuang, Biao Chen, Regan F. Volk, Katherine L. Hofmann, Balyn Zaro, Pao-Tien Chuang.

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
