## [Editor Report · Decision Letter 0]

28 Jan 2022

Dear Dr Chuang, 

Thank you for submitting your manuscript entitled "Wnt5a–Vangl1/2 signaling regulates the position and direction of lung branching through the cytoskeleton and focal adhesions" for consideration as a Short Reports by PLOS Biology.

Your manuscript has now been evaluated by the PLOS Biology editorial staff as well as by an academic editor with relevant expertise and I am writing to let you know that we would like to send your submission out for external peer review.

Once your full submission is complete, your paper will undergo a series of checks in preparation for peer review. Once your manuscript has passed the checks it will be sent out for review. To provide the metadata for your submission, please Login to Editorial Manager (https://www.editorialmanager.com/pbiology) within two working days, i.e. by Jan 31 2022 11:59PM.

If your manuscript has been previously reviewed at another journal, PLOS Biology is willing to work with those reviews in order to avoid re-starting the process. Submission of the previous reviews is entirely optional and our ability to use them effectively will depend on the willingness of the previous journal to confirm the content of the reports and share the reviewer identities. Please note that we reserve the right to invite additional reviewers if we consider that additional/independent reviewers are needed, although we aim to avoid this as far as possible. In our experience, working with previous reviews does save time. 

If you would like to send previous reviewer reports to us, please email me at ialvarez-garcia@plos.org to let me know, including the name of the previous journal and the manuscript ID the study was given, as well as attaching a point-by-point response to reviewers that details how you have or plan to address the reviewers' concerns. 

Given the disruptions resulting from the ongoing COVID-19 pandemic, please expect some delays in the editorial process. We apologise in advance for any inconvenience caused and will do our best to minimize impact as far as possible.

Kind regards,

Ines

--

Ines Alvarez-Garcia, PhD

Senior Editor

PLOS Biology

---

## [Decision Letter · Decision Letter 1]

13 Mar 2022

Dear Dr Chuang,

Thank you for submitting your manuscript entitled "Wnt5a–Vangl1/2 signaling regulates the position and direction of lung branching through the cytoskeleton and focal adhesions" for consideration as a Short Report at PLOS Biology. Thank you also for your patience as we completed our editorial process, and please accept my apologies for the delay in providing you with our decision. Your manuscript has been evaluated by the PLOS Biology editors, an Academic Editor with relevant expertise, and by three independent reviewers.

You will see that all reviewers find the conclusions of the manuscript interesting, however they also raise several concerns regarding the p-FAK staining, quantification of p-FAK in vivo, as well as the lack of a detailed cellular description, among others.

In light of the reviews (attached below), we will not be able to accept the current version of the manuscript, but we would welcome re-submission of a revised version that takes into account the reviewers' comments. We cannot make any decision about publication until we have seen the revised manuscript and your response to the reviewers' comments. Your revised manuscript is also likely to be sent for further evaluation by the reviewers.

We expect to receive your revised manuscript within 3 months. 

**IMPORTANT - SUBMITTING YOUR REVISION**

3. Resubmission Checklist

a) *PLOS Data Policy*

b) *Published Peer Review*

Sincerely,

Ines

--

Ines Alvarez-Garcia, PhD

Senior Editor

PLOS Biology

Reviewers' comments

Rev. 1:

In this manuscript from the lab of Pao-Tien Chuang, the authors report a very interesting lung branch positioning defect in Wnt5a knockout, mesenchymal Wnt5a deletion, epithelial Wnt5a deletion, and Vangl1/2 knockout mice, beginning around E11.5. They also report that subsequent branching defects were more severe in Wn5a deletion than with Vangl1/2 deletion, from which they conclude that Wnt5a has additional effects on canonical Wnt signaling including driving expression of Foxa2. Comparative bulk RNA sequencing with pathway analysis revealed perturbation in focal adhesion, ECM-receptor interactions, and actomyosin cytoskeleton, confirmed by proteomic analysis. They then localize p-FAK in wild type and mutant lungs by antibody staining and find it to be reduced in Vangl1/2 and Wnt5a mutant lungs, while proliferation is no different. They also use an FAK inhibitor in lung explant cultures which results in branch size and position defects. They propose that the Wnt5a-Vangl1/2 signaling axis regulates the position and direction of lung branching by regulating the cytoskeleton and focal adhesions. While the studies are of high quality and involve state of the art in vivo gene deletion with precision measurement of branching defects, it is not convincing that their model can explain the phenotype as they claim in their manuscript title.

Major criticisms:

1) It is hard to reconcile how perturbed cytoskeletal control and focal adhesions would result in mis-localized branches. I would rather expect truncated or failed branching defects at the normal site of initiation. It seems more likely that these mechanical defects would affect execution of branch outgrowth as opposed to specification of the site of branching.

2) Since Wnt5a deletion in either mesenchyme or epithelium appears to have the same result, and since p-FAK is active in both compartments in wild type lungs and reduced in both compartments in mutant lungs and with FAK inhibitor, it is not clear whether the phenotype can be ascribed to defective cytoskeletal control in the epithelial or mesenchymal compartment. In the Wnt5a knockout lung at E12.5, it even looks like p-FAK is only depleted in the mesenchyme (Figure S5 P,T), while the model suggests epithelial loss drives the phenotype. In all stains, there still appears to be p-FAK staining present in knockout lungs, so it is not clear that its absence underlies the phenotype.

3) There is incomplete penetrance in some genotypes which is not addressed. Perhaps the authors could examine p-FAK signaling in unaffected versus affected lungs from the same litter to see if this correlates, which would support their claim that defective focal adhesions are responsible for their phenotype.

4) In several important experiments, there was no statistical analysis performed (for example Figure 1, J-L and Figure 2, P-R).

Minor criticisms:

1) Data that is bi-modally distributed could be analyzed using a bi-modal statistical approach.

2) In some cases the wrong statistical test is applied. For example, in Figure 3T the authors should use a one-way ANOVA.

Rev. 2:

Zhang et al. investigated the mechanisms controlling the position and direction of lung branching in mice focusing on the early developmental stages. Using complex knockout studies, the authors show that Wnt5a controls branching morphogenesis via non-canonical PCP signaling, specifically Vangl 1/2. Moreover, the authors show that Foxa 2 is also involved in branching regulation. Interestingly, the authors show that perturbation of PCP proteins leads to changes in cellular organization and, specifically, the focal adhesion regulated by FAK. While I find the results highly relevant and that they can advance our understanding of lung morphogenesis, there are significant points that require to be addressed before this work is published. See below:

1. The authors claim that upon PCP perturbation, cytoskeletal components and mechanical cellular properties are impaired. However, the manuscript does not provide any detailed characterization of cellular morphology, measurements of cellular actomyosin properties, or ECM characterization in vivo. Without any more detailed cellular description, the model presented in Figure 4 T, U does not represent the manuscript's findings.

2. The authors should improve the quantification of p-FAK in vivo. As in point above, imaging at the cellular and sub-cellular levels would be essential to understand how p-FAK distribution changes.

3. It is unclear what the function of Foxa2 is in terms of regulating the morphology of brunching. Again, cellular imaging could help to distinguish between the role of Vangl and Foxa2

4. For FAK inhibitor experiments, please provide quantification of p-FAK reduction. Also, I can not find the figure of western blot mentioned in the text. Do levels of E-Cadhering also change upon FAK inhibition?

Minor comments:

1. For figure 1 J, K, L, statistical test is missing. Same for figure 2 P, Q, R,

2. Please consider changing the red/green combination to magenta/green

Rev. 3:

In this manuscript, the authors report that Wnt5a expressed in the lung mesenchyme, and to a lesser extent in the lung epithelium, acts through Vangl1/2 mediated PCP to regulate branching morphogenesis during early lung development. While the role of Wnt5a has been report in lung development, the authors made novel findings in this well written short report in the following aspects:

1. Identify the role of Wnt5a-Vangl1/2 PCP signaling in controlling the position and direction of early lung branching.

2. Showing that cytoskeletal reorganization and focal adhesion are regulated by Wnt5a-Vangl PCP in early lung morphogenesis.

3. Wnt5a regulates both PCP and Beta-catenin during lung morphogenesis.

The data are of high quality and nicely presented. Altogether the current study will spur future research in multiple directions.

Minor comments:

1.The authors should quantify p-FAK in Fig. 3E-N. The authors stated that "the levels of p-FAK were significantly reduced in the lung epithelium of mutant lungs, and this conclusion was confirmed by Western blot analysis of lysates from Wnt5a- or Vangl1/2-deficient lungs". I did not see any Western blotting results.

2. Fig. 3O-R, the authors showed cellular properties of control and Vangl1/2 knockout cells on fibronectin-coated dishes. What are these cells?

---

## [Decision Letter · Decision Letter 2]

4 Jul 2022

Dear Dr Chuang,

Thank you for your patience while we considered your revised manuscript entitled "Wnt5a–Vangl1/2 signaling regulates the position and direction of lung branching through the cytoskeleton and focal adhesions" for publication as a Short Report at PLOS Biology. This revised version of your manuscript has been evaluated by the PLOS Biology editors, the Academic Editor and two of the original reviewers.

Based on the reviews, we are likely to accept this manuscript for publication, provided you address the following data and other policy-related requests (see below).

We expect to receive your revised manuscript within two weeks. 

*Published Peer Review History*

*Press*

Sincerely,

Ines

--

Ines Alvarez-Garcia, PhD

Senior Editor

PLOS Biology

ETHICS STATEMENT:

Thank you for including the ethics statement. Please add the license/approval number used for animal care.

Fig. 1J, K, L, T; Fig. 2P, O, R, Y; Fig. 3A-D, K, B’, C’; Fig. S2G, N; Fig. S5G; Fig. S6A-C; Fig. S10M; Fig. S11Y, Z and Fig. S12Y

** Please also make sure that the RNAseq data you have deposited in GEO is made publicly available at this stage.

Reviewers' comments

Rev. 1:

In their revised manuscript, the authors have newly stained for p-cofilin and shown defects in Vangl1/2 mutant cell morphology, consistent with their model. They have also added a revised model for how branch position could be disrupted due to alterations in mechanical force. However, I still think direct experimental evidence supporting that cytoskeletal and focal adhesion defects are the basis for the mis-localized branch phenotype is lacking.

Wnt5a-Vangl1/2 are well-known to control the cytoskeleton and focal adhesions but how this translates into the observed phenotype remains rather vague and essentially purely speculative, which is my primary concern with this report.

Many of the proofs are quite indirect and the presence of p-FAK in the Wnt5a mutant lungs is still concerning that it may not be the primary effector of the phenotype, as implied in the manuscript title. The lack of a phenotype from isolated Vangl1/2 deletion in either the mesenchyme or epithelium does not clarify this concern. Similarly, the lack of information on whether the incomplete penetrance can be correlated with p-FAK levels leaves the picture confusing.

Overall, I feel the data still do not sufficiently support the main claim of the manuscript.

Rev. 2:

Dear Authors,

Thank you for addressing the comments. Levels of phosphorylated Cofilin (p-Cofilin) do not say much about actin filaments organisation/regulation. However, I agree with the authors that further, more detailed studies focusing on cytoskeletal mechanics are beyond the scope of this paper. Thus, in my opinion, the authors addressed and answered my comments adequately.

---

## [Editor Report · Decision Letter 3]

18 Jul 2022

Dear Dr Chuang,

Thank you for the submission of your revised Short Report entitled "Wnt5a–Vangl1/2 signaling regulates the position and direction of lung branching through the cytoskeleton and focal adhesions" for publication in PLOS Biology. On behalf of my colleagues and the Academic Editor, Emma Rawlins, I am happy to say that we can in principle accept your manuscript for publication, provided you address any remaining formatting and reporting issues. These will be detailed in an email you should receive within 2-3 business days from our colleagues in the journal operations team; no action is required from you until then. Please note that we will not be able to formally accept your manuscript and schedule it for publication until you have completed any requested changes.

PRESS

Sincerely, 

Ines

--

Ines Alvarez-Garcia, PhD, PhD

Senior Editor

PLOS Biology
